# Excitation and Reception of Higher-Order Guided Lamb Wave’s *A*_1_ and *S*_1_ Modes in Plastic and Composite Materials

**DOI:** 10.3390/ma15207249

**Published:** 2022-10-17

**Authors:** Rymantas Jonas Kazys, Justina Sestoke, Liudas Mazeika

**Affiliations:** Prof. K. Barsauskas Ultrasound Research Institute, Kaunas University of Technology, 51423 Kaunas, Lithuania

**Keywords:** air-coupled ultrasonic, Lamb waves, elastic properties, composite materials

## Abstract

**Featured Application:**

**The developed new method will allow use for more detailed analysis and characterization of defects that cannot be detected with *A*_0_ or *S*_0_ modes.**

**Abstract:**

Contemporary technologies are employing composite plate materials developed by using various innovative materials (nanostructures, mica structures, etc.). Application of higher-order modes could allow better detection and characterization of defects characteristic of planar plastic and composite structures, mainly due to shorter wavelength. However, excitation of higher-order modes meets many problems, especially in the case of the air-coupled technique, and is not sufficiently investigated. This is relevant in the cases of paper, high-density polyethylene (HDPE), membranes, GFRP, GLARE, CFRP and other composite structures. The objective of the paper was investigation of the excitation and reception of higher-order guided Lamb wave modes in plastic and composite plates. Therefore, it is appropriate to develop new non-contact ultrasonic measurement methods based on the excitation and reception of guided waves for the study of such objects. The obtained results clearly demonstrate the possibility to excite and receive efficiently different higher-order guided Lamb wave modes with very different phase velocities. The presented comparison of the experimental results with the simulation results showed a good agreement. The combination of air-coupled excitation and non-contact reception enables a non-destructive evaluation and characterization of moving plastic objects and composite structures.

## 1. Introduction

Contemporary technologies are employing composite plate materials developed by using various innovative materials (nanostructures, mica structures, etc.). Such materials are being used in the fields of electronics, specifically, in heating, lighting and electricity control devices and mechanics including the aerospace industry wherever exceptional physical properties are required, e.g., in compasses, rockets, oxygen breathing equipment, and hearing aids. In many cases, the manufacturing technologies do not allow employing contact measurement methods for quality control of the manufactured products [1,2,3]. However, ultrasonic methods enable investigation of key properties of materials that cannot be assessed by merely using traditional methods. For these purposes, ultrasonic guided waves propagating within the interior of the objects under investigation may be employed. The guided waves are already used for non-destructive testing and evaluation (NDT) of sheet-type materials. Usually, they are excited when the vibrations of an acoustic transducer are transferred to the investigated item by a direct contact via a narrow gap of liquid. However, contact liquids are impossible to use in many cases as the investigated item may be contaminated, damaged or otherwise negatively affected. This is relevant in the cases of paper, high-density polyethylene (HDPE), membranes, GFRP, GLARE, CFRP and other composite structures [4,5]. 

The guided waves in sheet-type materials may be excited via an air gap in two different ways depending on the ratio of propagation velocities of ultrasonic waves in the material and air [6,7]. When the ultrasound velocity in air is lower than the phase speed of the guided waves in the object, the optimal incident angle of the ultrasonic wave propagating from the air onto the plate exists at which the biggest amplitude of the excited guided wave is obtained. Then, the propagating guided wave in the plate material radiates a leaky wave into air which may be used for air-coupled reception [8]. This mode is commonly applied for detecting defects inside composite materials as the leakage is increasing at the defective zone. The attenuation of acoustic waves in air and in the object under investigation increases with the frequency. The losses of ultrasonic signals may be reduced by generating guided waves while using lower frequencies. We used novel air-coupled ultrasonic techniques based on the excitation of guided waves or by phased arrays at lower frequencies, or by deflected ultrasonic transducers at higher frequencies depending if the phase velocity of the mode is higher or lower than the velocity in air. It should be remarked that both methods of the excitation and reception of guided waves are innovative; neither of them has been explored yet [9,10].

Usually, for non-destructive evaluation, zero-order Lamb wave antisymmetric A_0_ and symmetric S_0_ modes are used [4,11,12]. However, application of higher-order modes could allow better detection and characterization of defects characteristic of planar plastic and composite structures. There are already some publications describing application of higher-order guided waves for detection of wall thickness reduction and crack-like defects, however only in metallic structures [13,14,15,16]. For example, P. Khalili and P. Cawley analyzed symmetric S_0_ mode at ~1.5 MHz-mm and shear horizontal SH_0_ and SH_1_ modes at ~3 MHz-mm in the 10 mm thick steel plate and determined that they can be suitable for detection of shallow defects [15]. Authors for that exploited SH_1_ waves, but parasitic SH_0_ was excited as well, and this can complicate the inspection process and, therefore, limit defect detectability. Authors showed that the SH_1_ mode was sensitive to gradual and smaller defects down to 10% of wall thickness loss [15]. In another paper, P. Khalili and F. Cegla presented research focused on application of the SH_1_ wave for inspection of the 10 mm thick aluminum plate with crack-like defects [16]. By means of analytical predictions and finite element modeling, it was shown that a signal dominated by the SH_1_ mode can be generated using a single electromagnetic acoustic transducer with a permanent periodic magnet. It was also shown that, by studying the reflection coefficient of the SH_1_ mode, the pure SH_1_ mode can be used to detect defects as shallow as a 5% thickness loss from a 500 mm standoff. Other authors in their articles examined the spectra of acoustic oscillations generated by interdigital transducers in a plate made of LiNbO_3_ piezo crystals. They found that along with the zeroth and higher-order modes, this spectrum also contains odd harmonics of the same modes. When the velocities are close, the amplitude–frequency response modes and harmonics can be too close and even interfere with each other, distorting the entire spectrum of acoustic vibrations. The authors suggested that plates up to three wavelengths thick and transducers with bandwidths up to 5% and operating frequencies below the first harmonic are most suitable for practical use [17,18]. Other authors studied the influence of thin interlayers on the behavior of elastic guided waves in isotropic laminate structures, paying particular attention to the application of the obtained results to determine the mechanical properties of such sublayers and to the evaluation of the integrity of the interlayer contact using a laser Doppler vibrometer [19,20,21,22]. The authors claimed that the use of effective boundary conditions can explain the observed behavior of high-order elastic guided waves in the considered laminate structures (i.e., emergence of mode pairs, mapping of closed-form boundary frequencies, etc.). Unlike these authors, our aim was to apply ultrasonic air-coupled transducers to excite and obtain higher-order Lamb wave modes.

The presented examples deal with non-destructive testing of metallic and ceramics samples, elastic properties of which are very different from other plastic and composite materials. Excitation and reception of higher-order modes in such materials also should be different. The excitation process depends on the spatial distribution of the surface displacements of the sample, which should be excited by an ultrasonic transducer.

Symmetric modes possess two displacement components: vertical and longitudinal. The amplitude of the longitudinal component at low frequencies is many times greater than the amplitude of the vertical component on the surface of the sample. It means that excitation of symmetric modes is also more complicated than antisymmetric modes in composite materials. Asymmetric modes are more versatile because the amplitudes of the vertical and longitudinal displacement components are of a similar order. It allows exciting such modes using both methods: the contact and the air-coupled. 

In the case of the air-coupled excitation, it is necessary to find the frequency range and the angle of incidence of the ultrasonic wave in air, at which the required guided wave mode in the plastic sample is optimally excited. Those parameters can be determined from dispersion curves of guided waves propagating in the object under test. In most cases, the dispersion curves for particular materials are not known in advance, and therefore they should be calculated or measured beforehand. For that, elastic constant of the materials is necessary to know, which usually is found from the measured phase velocities in investigated plastic or composite materials [23,24,25,26,27,28]. It is necessary to point out that in the case of materials with high attenuation, phase velocities are frequency-dependent even in bulk specimens, and therefore methods based on reconstruction of the phase velocity dispersion curves should be used [17,18]. In our case, attenuation is not very high, and therefore the elastic constants of the materials were determined using methods described in [25,26,27,28]. 

However, until now, majority of publications have been devoted to the analysis of application of zero-order antisymmetric modes for detection of various defects in composite, metal and plastic materials. Application of the higher-order Lamb wave modes through an air gap is very limited because their selective excitation and reception are much more complicated. For this purpose, air-coupled ultrasonic transducers can be applied; however, at the moment, they are mainly exploited for the excitation and reception of the lowest Lamb wave modes. Thus, there is a lack of research demonstrating excitation of high-order asymmetric and symmetric modes for defect detection in various composite materials.

The objective of the paper is investigation of the excitation and reception of higher-order guided Lamb wave modes in plastic and composite plates. 

The paper is organized as follows. Section 2 reviews the properties of higher-order guided wave modes and presents the theoretical background. The experimental investigations performed by contacts and air-coupled methods are reported in Section 3. Conclusions and discussion can be found in Section 4.

## 2. Properties of Higher-Order Guided Wave Modes

To analyze the physics of higher-order guided Lamb wave modes, we chose high-density polyethylene (HDPE). It is prepared from ethylene by a catalytic process [10,29]. It possesses the absence of branching resulting in a more closely packed structure with a higher density and somewhat higher chemical resistance than LDPE. Compared to engineering plastics, it has lower thermal and mechanical properties, such as tensile stress and flexural and compressive strength. Compared to high-molecular-weight PE, it is more rigid, and its resistance to continuous shock is therefore lower [30,31].

PE-300 is also somewhat harder and opaquer, and it can withstand rather higher temperatures (120° Celsius for short periods, 110° Celsius continuously). High-density polyethylene lends itself particularly well to blow molding. HDPE Polyethylene 300 has the following properties: rigidity, resilience, durability, chemical and moisture tolerance, gas permeability, ease of processing and ease of formation. PE-300 plastic is used for wastewater treatment plants, for production of various tanks, water slides, petroleum and chemical receptacles, shelves, and partitions and for the bottling industry, including production of snow arena guard rails, playgrounds, supporting/guard bars, security systems for motorcycles, quadricycles, motor vehicles, etc. It is easy to weld or form by heat treatment. The low friction coefficient and its non-hygroscopicity make it suitable for use as a bearing or mechanical parts with low loads, even when operating in water [32,33,34]. 

To know the distributions of higher-order modes in those materials, one must first know their exact viscoelastic properties, which were measured at the Physical Acoustics department (APY), Bordeaux, France. Dimensions of the sample selected for measurements were 200 × 200 mm. The experimental equipment and set-up are presented in Figure 1 and Figure 2. The experimental set-up consists of a LeCroy 9450A Dual 300 kHz oscilloscope (LeCroy, Geneva, Switzerland), Olympus 5077PR square wave pulse/receiver (Bucharest, Romania), Newport motion controller model MM4006 (Champaign, IL, USA), V 305 2.25 MHz/0.75 111,280 unfocussed immersion transmitter ultrasonic transducer (Sofranel, Sartrouville, France) and V 305 2.25 MHz/0.75″ 11,282 receiver ultrasonic transducer (Sofranel, Sartrouville, France). The diameter of the ultrasonic transducers is 0.75″, receiver attenuation is −40 dB, and excitation voltage is *U_ex_* = 100 V. Two different materials were experimentally investigated: high-density polyethylene (HDPE) and glass fiber (GFRP) composite material. Before measuring the viscoelastic properties, dimensions of the samples were measured, and the volumes of the samples were calculated. The samples were weighed on a scale: the weight of the polyethylene (HDPE) sample is 237.30 g, and the weight of the glass fiber (GFRP) sample is 407.31 g. The sample densities were calculated using the measurement data. The measured viscoelastic properties and elastic parameters of the HDPE sample are presented in Table 1 and Table 2. The calculated densities are given in the corresponding tables (Table 2 and Table 4). The viscoelastic properties of HDPE and GFRP material were measured in the water tank, the water temperature was 23 °C, and scanning was performed from 0° to 72° by step 2°. The GFRP material was measured using 1 MHz ultrasonic transducers. The features and appearance of the ultrasonic transducers are the same as the 2.25 MHz ultrasonic transducers. The used measurement method is widely known and described in the following articles [25,26,27,28,33,35].

The dispersion curves were calculated using two different methods: the semi-analytical finite element (SAFE) method (Institute of Ultrasound, Kaunas, Lithuania) and the PROPAG software (APY Department, Bordeaux, France). The dispersion curves were verified by experimental investigations. They were carried out on a PE-300 specimen with clearly expressed properties, which were measured in the APY department. In this sample, the antisymmetric modes A_0_ and A_1_ and symmetric S_0_ and S_1_ modes were excited and signals recorded; the measurement investigations of the PE-300 sample were performed by the ULTRALAB system (Institute of Ultrasound, Kaunas) using the experimental set-up presented in the previous section. The experimental set-up was the same as used for investigation of the PE-300 sample and measurements for glass fiber (GFRP) composite material. Pultruded glass fiber reinforced polymer (GFRP) has light weight, good strength and excellent resistance to corrosion [36,37]. The GFRP material was measured using 1 MHz ultrasonic transducers. The measured viscoelastic properties and elastic parameters of the HDPE sample are presented in Table 1 and Table 2. The measured viscoelastic properties and elastic parameters of the GFRP composite material are presented in Table 3 and Table 4. Young’s modulus and Poisson’s ratio are given with the accuracy of 10^−4^ to compare the two different calculation methods of the dispersion curves. Note that this accuracy is only used for calculation to get the best possible match of results.

The measurements showed that the PE-300 sample is an isometric material, and then the other viscoelastic properties are C_22_ = C_11_, C_33_ = C_11_, C_44_ = C_66_, C_55_ = C_66_, C_12_ = C_11_ − 2 × C_66_, C_13_ = C_12_, C_23_ = C_12_. The elastic parameters of the PE-300 sample are presented in Table 2. The elastic parameters were calculated of the measured viscoelastic properties in the HDPE sample. From the measured results follows that HDPE material is isotropic material and GFRP material is anisotropic material. Knowing the exact elastic properties of the material, we can calculate the phase and group velocities of the modes and the mode distributions. 

The calculations were performed by two different methods: using the semi-analytical finite element (SAFE) method (Ultrasound Institute, Kaunas) and compared with the simulated dispersion curves in the APY department (Bordeaux, France). The simulated dispersion curves of the phase and group velocities, for the 4 mm HDPE plate, are shown in Figure 3. In Figure 3a are presented phase velocities and in Figure 3b group velocities of various modes. The black solid lines show velocities simulated by the SAFE method in the Ultrasound Institute, Kaunas, and the various color dots indicate velocities obtained in the APY department, Bordeaux, using the PROPAG software [27,38,39,40]. The presented results demonstrate a very good correspondence of the results obtained by two different methods.

The dispersion curves of the Lamb waves were calculated using the semi-analytical finite element (SAFE) method [41]. The SAFE method can be used not only for isotropic materials but also for layered anisotropic materials. The dispersion curves of the phase and group velocities, for the 4 mm GFRP plate, are shown in Figure 4, respectively. The direction used in calculation by the SAFE method for anisotropic GFRP material was XY.

Knowing the phase velocities of the sample from the dispersion curves, it is possible to determine at what angle of incidence the higher-order modes should be excited, for example, the antisymmetric mode A_1_ and symmetric mode S_1_. Let us take a closer look at the HDPE sample. Usually, for air-coupled excitation and reception of the antisymmetric Lamb wave modes, ultrasonic transducers are oriented to the plate structure under test at the optimum incidence angle *θi*. There is the frequency-dependent phase velocity of the guided Lamb wave modes in the structure under a sample. The value of the optimum incidence angle *θi* according to Snell’s law (Equation (1)) depends on the ratio of the phase velocities *V_air_/V_ph_(f)*. For the case when the ultrasound phase velocity in the structure is greater than or equal to the sound velocity in air *V_ph_(f)* ≥ *V_air_*, the incidence angle *θi* can be calculated as shown in Equation (1) [36]:*θi* = arcsin (V*_air_*/V*_ph_ (f)*)(1)
where *V_ph_* is the phase velocity of the selected mode in the HDPE plate found from the dispersion curves, and *V_air_* is the velocity of sound in air. The ultrasound velocity in air at 21 °C was 343.21 m/s. From the dispersion curves, the following values were obtained for the 4 mm HDPE plate at 300 kHz excitation frequency: the phase velocity of *A*_1_ mode is 1508 m/s, and the group velocity is 756 m/s. Using the phase velocity, the angle of incidence of *θ_i_* = 22.96° was calculated using Snell’s law. The wavelength in air at 300 kHz was 1.4 mm, the wavelength of the *A*_0_ mode in the HDPE plate at 300 kHz was 2.93 mm, the wavelength of the *A*_1_ mode was 5.02 mm, and the wavelength of the *S*_0_ mode was 3.12 mm. The distributions of particle displacements across the 4 mm thickness of the sample at the 300 kHz of the antisymmetric modes *A*_0_ and *A*_1_ and symmetric *S*_0_ mode are presented in Figure 5, Figure 6 and Figure 7. The amplitude values in Figure 7e are normalized with respect to the maximum displacement values.

The antisymmetric modes A_0_ and A_1_ and symmetric S_0_ mode distributions of particle displacements across the 4 mm thickness of the sample by 300 kHz are presented in Figure 5, Figure 6 and Figure 7. From the presented results follows that in those modes, there are two components: vertical and longitudinal (tangential). In this paper, we present both components. The normal component of the antisymmetric modes *A*_0_ and *A*_1_ has the same sign on both sides, with a large amplitude on the surface. The normal component of the symmetrical *S*_0_ mode has opposite signs on both sides and a large amplitude on the surface. Due to this fact, it can be assumed that at 300 kHz frequency, the *A*_0_, *A*_1_ and *S*_0_ modes can be excited through the air gap with ultrasonic air-coupled transducers. In the next Section 3, we present the experimental investigation performed by two different methods: contact and air-coupled methods.

## 3. Experimental Investigation

The purpose of the experimental investigation presented in this section was to check whether the calculated dispersion curves correspond to the experimental ones and to check how it is possible to excite and receive the corresponding higher-order Lamb wave modes. Experimental investigations were performed by two different methods: the contact method using contact ultrasonic transducers and with the air-coupled ultrasonic transducers.

### 3.1. Contact Method

Experimental investigations were performed using the ultrasonic equipment and set-up shown in Figure 8. It consists of the ultrasonic measurement system ULTRALAB with a linear scanner (Ultrasound Institute of Kaunas University of Technology), an amplifier and contact ultrasonic transducers [42,43,44]. These are low-frequency broadband contact type ultrasonic transducers operating in the frequency range of 50 kHz to 350 kHz. The experimental investigations were carried out with a convex protector with the diameter of 5 mm and thickness of 4 mm, and the diameter of the contact area was 0.5 mm. The acoustic contact between the transmitter and the receiver was obtained by a coupling liquid (glycerol) which was used to obtain a more stable acoustic contact with the ultrasonic transducers during scanning. It allowed obtaining better stability of the ultrasonic signals and correspondingly obtaining a higher accuracy of measurements. The transmitting transducer was excited by the signal for *A*_1_ mode *f_A_*_1_ = 170 kHz. The sampling frequency was set to 100 MHz. Dimensions of the sample selected for measurement were 700 × 700 mm. The distance between the transmitter and receiver transducers was set to 80 mm. The scanning distance was set to 150 mm from the receiver; the step was correspondingly 0.1 mm.

In this case, the excitation by the 3-period burst with the amplitude of 170 V was used. The measured B-scan obtained in the case of the contact excitation of a 4 mm thick PE-300 sample is presented in Figure 9, and the received signal at the start position is presented in Figure 10. The amplitude of the values in Figure 10 is presented in arbitrary units (a.u.).

For determination of the dispersion curves for the PE-300 sample under investigation, we used the method based on the analysis of a 2D spatial-temporal spectrum of the B-scan of the normal displacements [45]. Dispersion curves using the semi-analytical finite element method (SAFE) are shown by red lines, and the 2D Fourier transform of the B-scan converted in the frequency-phased velocity domain is presented in Figure 11. Lamb waves propagating at different velocities are indicated in this figure. More about the 2D Fourier transform and 2D spatial-temporal spectrum of the measured B-scan methods can be found in our paper [45]. Using the 2D spatial-temporal spectrum of the measured B-scan modes in the case of the PE-300 sample, where the red dotted line is A_1_ mode from the simulated SAFE method, is presented in Figure 12. It can be seen from the presented spectrum that it is possible to recognize and identify modes from Lamb waves propagating at different velocities. The color scale of the amplitudes is shown on the right side of the image; the amplitudes of the values are presented in arbitrary units. The filtered and normalized amplitude of the antisymmetric A_1_ mode is presented in Figure 13. The amplitudes of the values in Figure 13b are normalized with respect to the maximum displacement values. 

The dispersion curves calculated by the semi-analytical finite element method (SAFE) shown by the solid red lines and the 2D Fourier transform of the measured B-scan data are presented in Figure 11. The presented results demonstrate a very good correspondence of the results obtained by two different methods: the first simulated by SAFE and the second measured and processed by the FFT_2D_. The obtained results demonstrate feasibility to excite *A*_0_, *A*_1_, *A*_2_ and *S*_0_ waves (Figure 12) using the ultrasonic contact transducers operating in the frequency range of 120 kHz to 200 kHz. The measurement results also show that the maximum displacement amplitude of the sample surface is obtained in the case of the symmetric S_0_ mode. The displacement amplitude of the antisymmetric *A*_1_ mode is 3 times lower than that of the symmetric *S*_0_ mode (Figure 13a). In spite of that, the normal component of the A_1_ mode has a sufficiently large amplitude on the surface, indicating the possibility to excite this mode through the air gap. The possibility to excite the antisymmetric A_1_ guided wave mode by the air-coupled method is discussed in the next Section 3.2.

### 3.2. Air-Coupled Method

Other experiments were performed using the air-coupled method. The PE-300 specimen was investigated using one pair of air-coupled ultrasonic transducers manufactured at the Ultrasound Institute (UI) of Kaunas University of Technology (Kaunas, Lithuania) [26,27,28]. The pair consisted of single-element unfocused air-coupled transducers with 300 kHz operating frequency. The transducers used in this investigation are shown in Figure 14a. The active transducer diameter for the flat transducer was 14 mm, the transduction loss per transducer was −28 dB, and the relative bandwidth (Δ*f*/*f*_0_) was 53%.

The received signal when transmitting and the receiving transducers operating “face to face” through an air gap are presented in Figure 15. The amplitude of the signal in Figure 15 is presented in arbitrary units (a.u.). The measurement conditions were the following: excitation with a 3-period burst, the excitation voltage *U_ex_* = 150 V, the frequency was 300 kHz, and the distance between the transducers was *d* = 60 mm. The experiment was performed by rotating the sample in a vertical position in the angular sector from −45° to 45°. The rotation step was 0.1°. 

The measured B-scan obtained in the case of the air-coupled excitation of a 4 mm thick PE-300 sample is presented in Figure 16, and the amplitudes of the measured B-scan are shown in Figure 17. The amplitude in Figure 17 is presented in arbitrary units (a.u.). From the measurement results follows that higher-order mode is excited in the HDPE specimen, e.g., the antisymmetric mode *A*_1_ and symmetrical *S*_1_ mode.

The aim of this measurement was identification of higher-order modes. From the measurement results follows that those two higher-order modes—the antisymmetric mode *A*_1_ and symmetric *S*_1_ mode—are excited. The higher-order modes were identified by the angle of incidence, which was calculated using Snell’s law. The experiment was performed after turning the sample at an angle of 70° and scanning in the Y = 70 mm direction. The scanning and rotation steps were correspondingly 0.1° and 0.1 mm. The distance between the transmitter and receiver transducers was set at 45 mm, the excitation by the burst of 3 periods, the excitation voltage *U_ex_* = 650 V and the frequency *f* = 300 kHz. From the dispersion curves, the following values were obtained for the 4 mm HDPE plate at 300 kHz excitation frequency: the phase velocity of the *A*_1_ guided wave is 1508 m/s and the group velocity 756 m/s. Using the phase velocity, the angle of incidence required to excite the *A*_1_ guided wave was calculated using Snell’s law and is *θi* = 22.96°. The phase velocity of the *S*_1_ guided wave is 2046 m/s, and the required angle of incidence is *θi* = 16.85°. The measured B-scan obtained in the case of the air-coupled excitation of a 4 mm thick PE-300 sample is presented in Figure 18, and the amplitude of the measured B-scan is shown in Figure 19. 

From the air-coupled measured B-scan and amplitude images (Figure 18 and Figure 19), it follows that according to the identification criterion, e.g., the angle of incidence, two Lamb wave modes are excited. One mode is at the angle of incidence ~*θi* = 22° and another mode at ~*θi* = 16°. From the presented results of the air-coupled measurements, we can conclude that two higher-order modes are excited and received in the plastic specimen, e.g., the antisymmetric mode A_1_ and the symmetric mode S_1_. Such excitation method can be used in the case of other complex composite materials as well.

## 4. Conclusions and Discussion

Various modes of guided waves are already used for non-destructive testing and evaluation of composite structures. It is possible to expect that higher-order symmetric and antisymmetric modes may improve detection and characterization of various defects in plastic and composite materials. For that, first of all, efficient techniques for the excitation and reception of such modes are necessary. The aim of this work was to study the excitation and reception of higher-order guided Lamb wave modes in plastic and composite plates. For excitation of such modes, it is necessary to select the proper frequencies, which depend on the mode type, thickness of the sample and elastic properties of the material. If all those parameters are known, then the necessary frequencies can be found from the dispersion curves. However, in many cases, elastic properties of the material are not known with the necessary accuracy; therefore, we proposed at the very beginning to measure elastic constants and to use them for the calculation of accurate dispersion curves. Viscoelastic properties were measured at the Department of Physical Acoustics (APY), Bordeaux, France. The dimensions of the plastic and composite samples selected for measurements were 200 × 200 mm. Dispersion curves were calculated using two different methods: the semi-analytical finite element (SAFE) method (Ultrasound Institute, Kaunas) and the PROPAG software (APY department, Bordeaux). The theoretical calculations were performed using the semi-analytical finite element method, which made it possible to find out the frequency ranges in which higher-order modes can be excited. Theoretical calculations showed that it is possible to excite higher-order modes at 300 kHz. The dispersion curves also allow selected frequencies to determine phase and group velocities of particular modes.

To excite and receive efficiently higher-order modes through an air gap, the ultrasonic wave in air must be directed to the sample at the angle of incident which depends on the phase velocity of the mode to be excited. It is necessary to keep in mind that phase velocities depend not only on the mode type but also on the frequency used to excite the required mode. Those velocities are found from the calculated (or measured) dispersion curves and are used to determine the deflection angle of the air-coupled ultrasonic transducer to get the necessary angle of the incidence. 

It is necessary to point out that in the frequency range where higher-order modes may exist, usually not a single but a few different modes may be excited. It may complicate identification of what modes actually were excited. For identification, we proposed the following identification criteria: the phase velocity *c_ph_* and the angle of incidence *θ_i_* at which the maximal amplitude of the required mode is obtained. 

During experimental investigations, we checked whether the calculated dispersion curves correspond to the experimental ones and found a good agreement. Those dispersion curves were exploited for determination of the frequencies and incident angles necessary for the excitation and reception of the required higher-order Lamb wave modes. The experimental investigations were performed by two different methods: a contact method using point-type ultrasonic transducers with 0.1 mm aperture and an air-coupled method using air-coupled ultrasonic transducers. Higher-order guided Lamb wave modes were excited and received by the contact ultrasonic transducers operating in the frequency range from 50 kHz to 350 kHz. The amplitudes of higher-order guided Lamb wave modes were measured by the ULTRALAB system at different distances from the excitation region, and in such a way, a B-scan was obtained. The phase velocities were determined from the B-scan applying a 2D spatial and temporal spectrum analysis. The experimental results obtained by the contact method demonstrate that it is feasible to excite higher-order A_1_ and S_1_ modes in plastic and composite materials. 

For practical applications, more attractive is the air-coupled method; therefore, other experiments directed to the excitation and reception of higher-order modes were carried out using this method. Experiments were performed by one pair of air-coupled ultrasonic transducers manufactured at the Ultrasound Institute of Kaunas University of Technology (UI, KTU). The pair consisted of single-element unfocused air-coupled transducers with an operation frequency of 300 kHz. The obtained results show that two higher-order modes, such as the antisymmetric mode A_1_ and the symmetric mode S_1_, were excited. Higher-order modes were identified by the angle of incidence, which was calculated using Snell’s law and the phase and group velocities of the excited modes. In the investigated plastic HDPE plate with 4 mm thickness at the excitation frequency of 300 kHz, the following higher-order A_1_ and S_1_ modes were observed. The antisymmetric A_1_ mode was excited by the air-coupled ultrasonic wave incident at the angle *θ_i_* = 22.96°. The phase velocity of the excited A_1_ guided wave was 1508 m/s. Correspondingly, the symmetric S_1_ mode was excited at the angle of incidence *θ_i_* = 16.85°. The phase velocity of the guided S_1_ wave was 2046 m/s. The measured velocities correspond well to the velocities of the corresponding modes in the calculated dispersion curves. 

The obtained results clearly demonstrate the possibility to excite and receive efficiently different higher-order guided Lamb wave modes with very different phase velocities. The presented comparison of the experimental results with the simulation results showed a good agreement. The combination of air-coupled excitation and non-contact reception enables a non-destructive evaluation and characterization of moving plastic objects and composite structures. In the future, it is planned to perform measurements in samples of complex geometry to identify and investigate various defects in such specimens using air-coupled phased arrays with novel piezoelectric materials.

## Figures and Tables

**Figure 1 materials-15-07249-f001:**
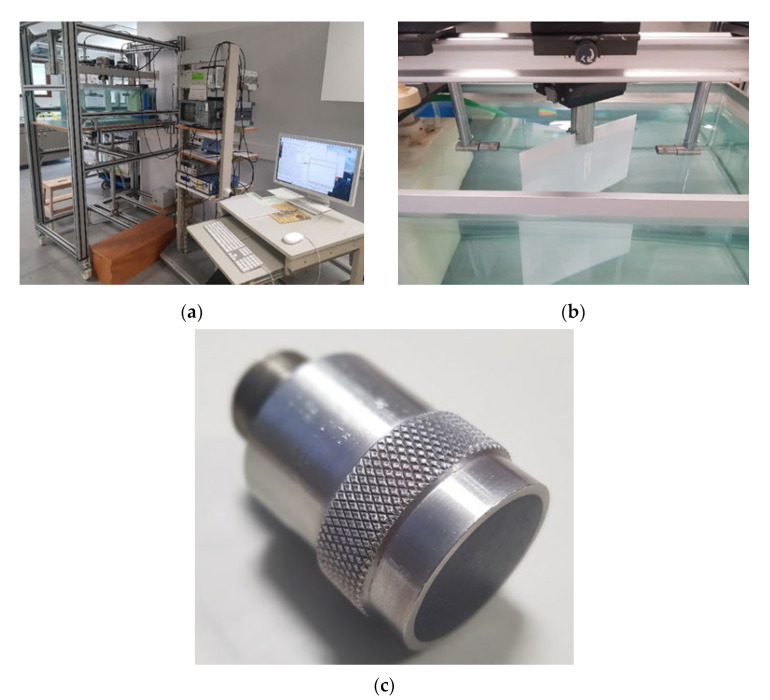
(**a**) Equipment of the measurement; (**b**) PE-300 sample; (**c**) V 305 2.25 MHz/0.75 unfocussed immersion ultrasonic transducer.

**Figure 2 materials-15-07249-f002:**
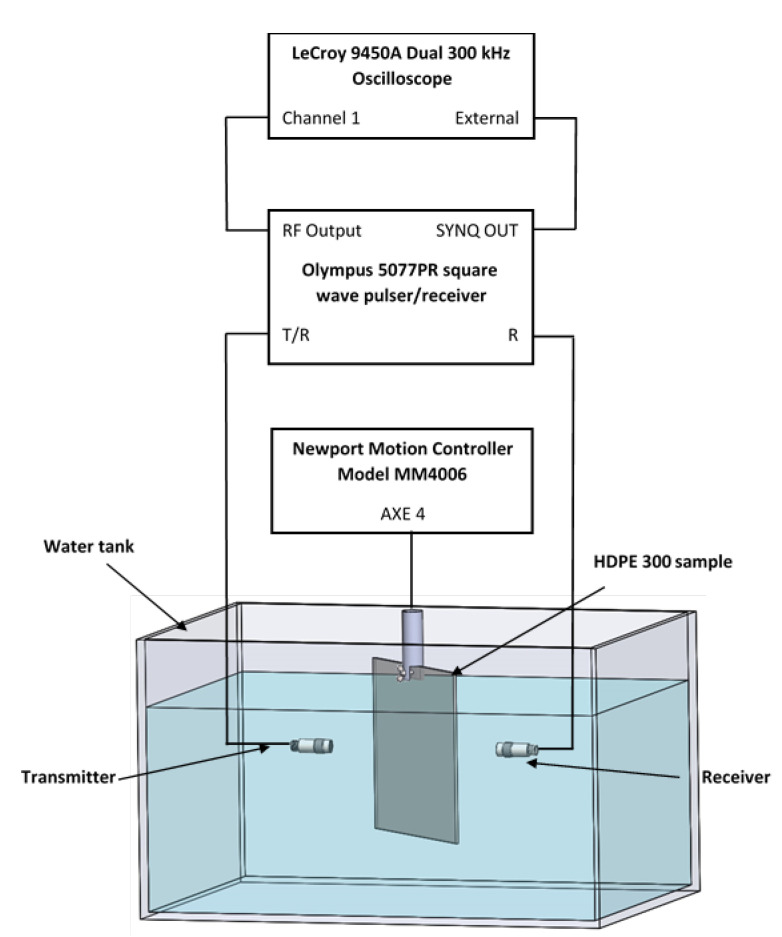
Experimental set-up with immersion ultrasonic transducer in the water tank.

**Figure 3 materials-15-07249-f003:**
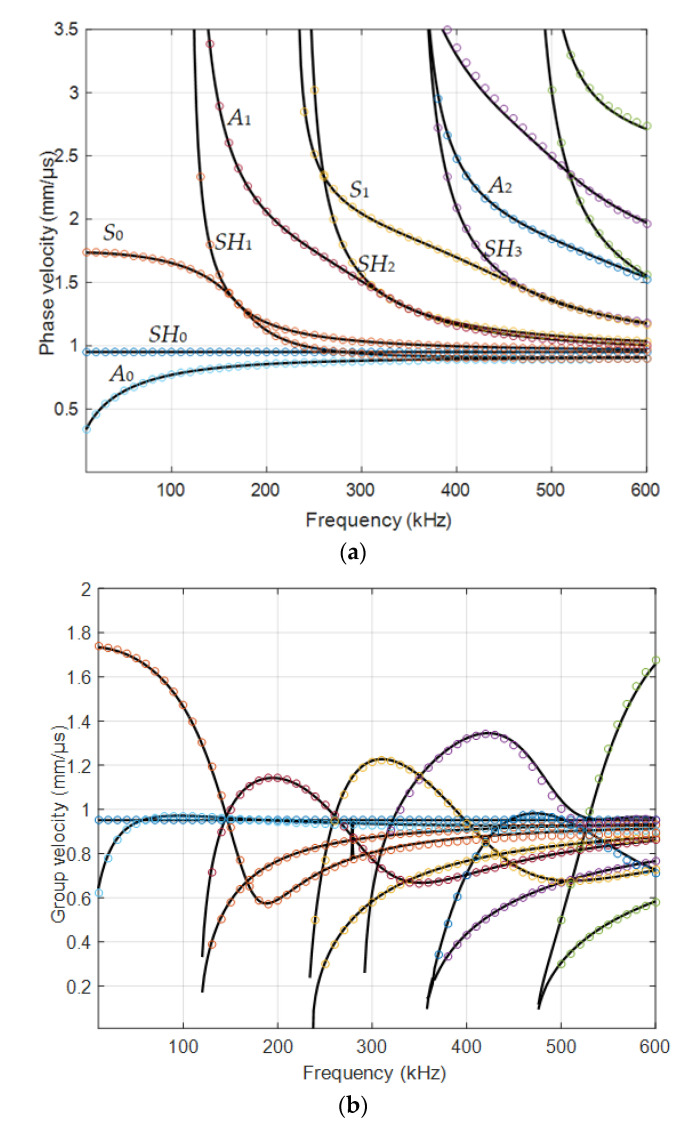
Simulated dispersion curves of the HDPE 4 mm sample: (**a**) phase velocities; (**b**) group velocities. Black solid lines—simulated by the SAFE method in the Ultrasound Institute, Kaunas; various color dots—APY department, Bordeaux, using PROPAG software.

**Figure 4 materials-15-07249-f004:**
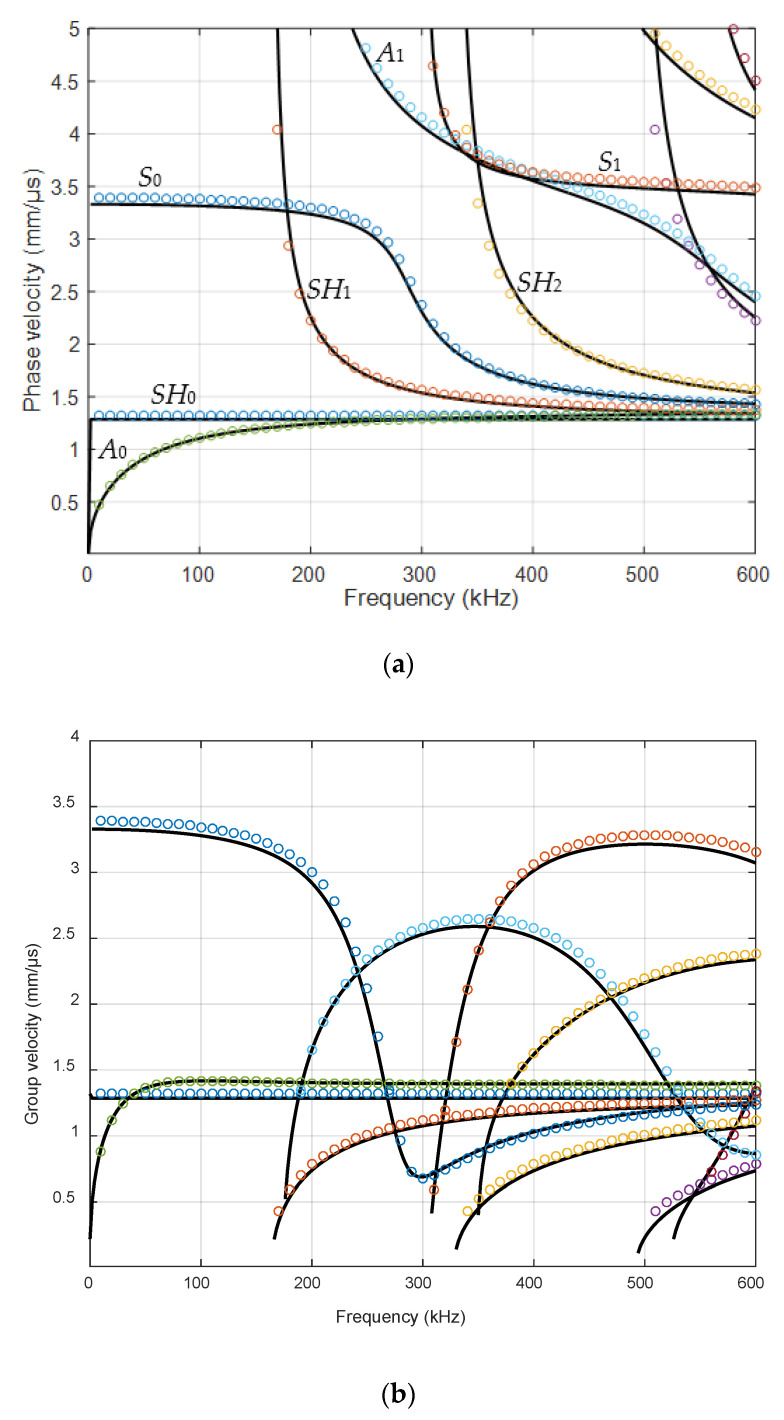
Simulated dispersion curves of the GFRP 4 mm sample: (**a**) phase velocities; (**b**) group velocities. Black solid lines—simulated by the SAFE method in the Ultrasound Institute, Kaunas; various color dots—APY department, Bordeaux, using PROPAG software.

**Figure 5 materials-15-07249-f005:**
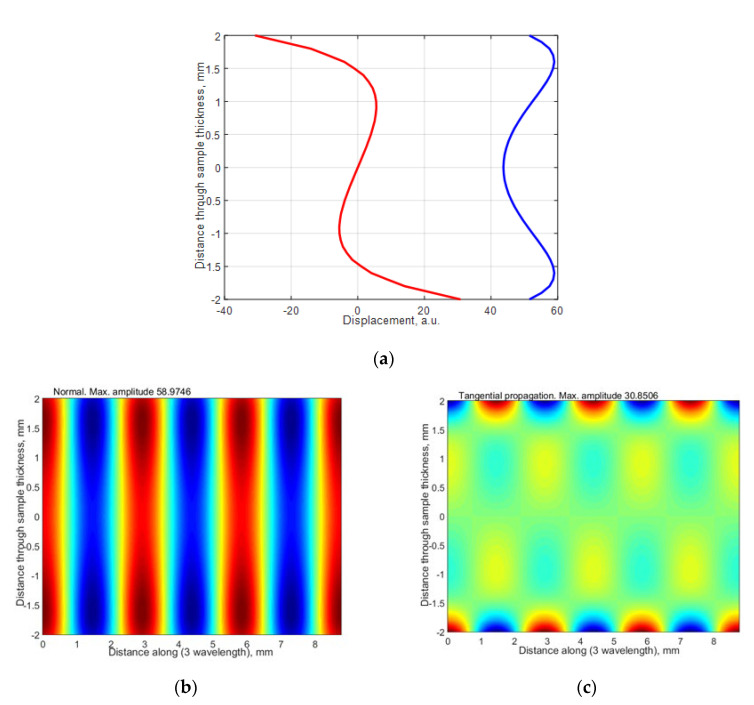
The distribution of particle displacement of antisymmetric A_0_ mode across 4 mm thickness of sample by 300 kHz: (**a**) the blue color is a normal displacement, the red color is a tangential displacement; (**b**,**c**) 3 wavelengths of the mode are given.

**Figure 6 materials-15-07249-f006:**
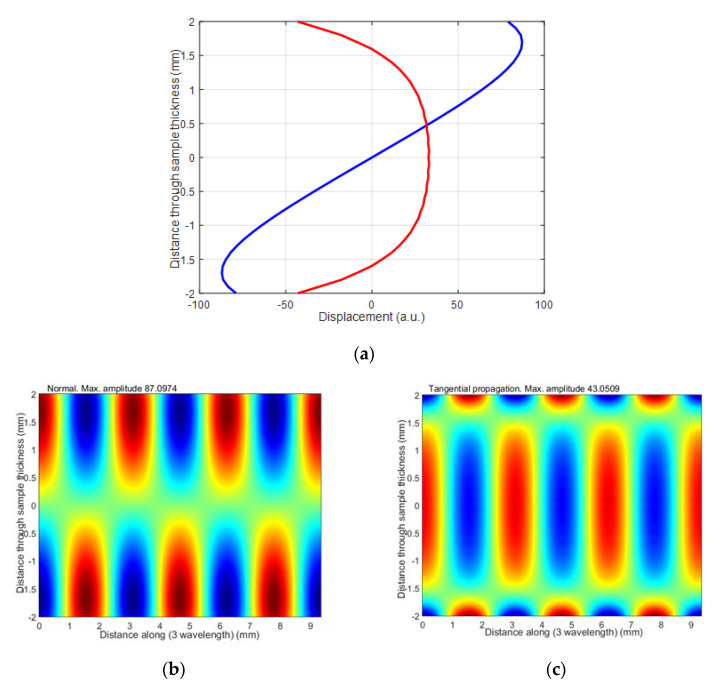
The distribution of particle displacement of antisymmetric S_0_ across 4 mm thickness of sample by 300 kHz: (**a**) blue color is normal displacement, red color is tangential displacement; (**b**,**c**) 3 wavelengths of the mode are given.

**Figure 7 materials-15-07249-f007:**
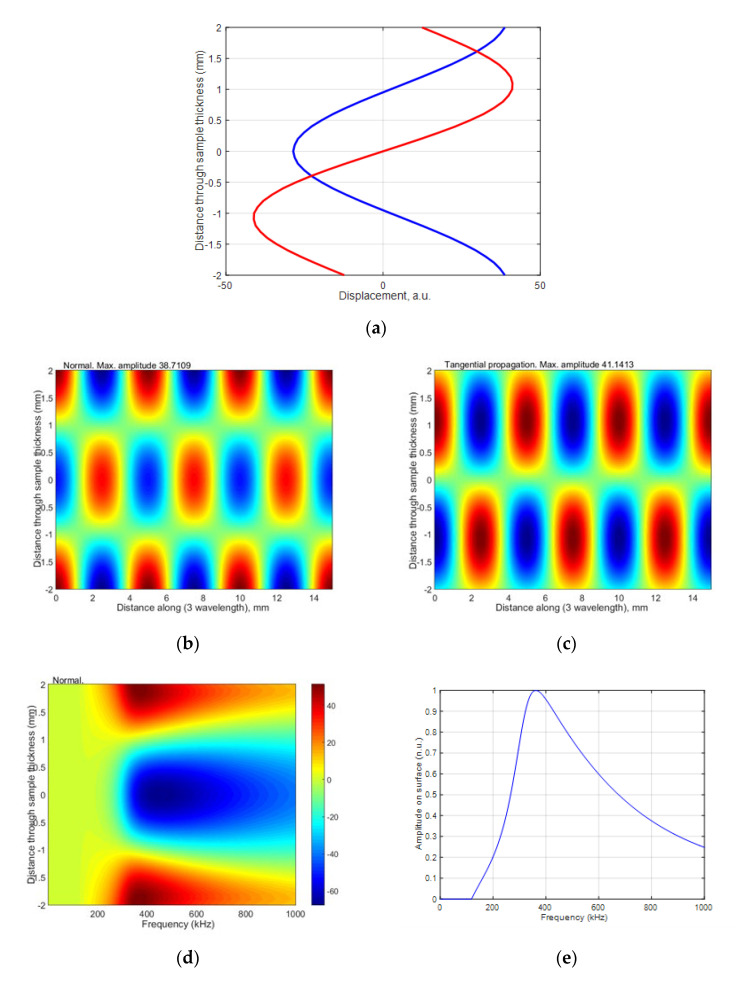
The distribution of particle displacement of antisymmetric A_1_ mode across 4 mm thickness of sample by 300 kHz: (**a**) the blue color is a normal displacement, the red color is a tangential displacement; (**b**,**c**) 3 wavelengths of the mode are given; (**d**,**f**) normal and tangential displacements from the frequency-dependent; (**e**,**g**) normalized displacement amplitudes on the surface of the sample ((**e**) normal, (**g**) tangential).

**Figure 8 materials-15-07249-f008:**
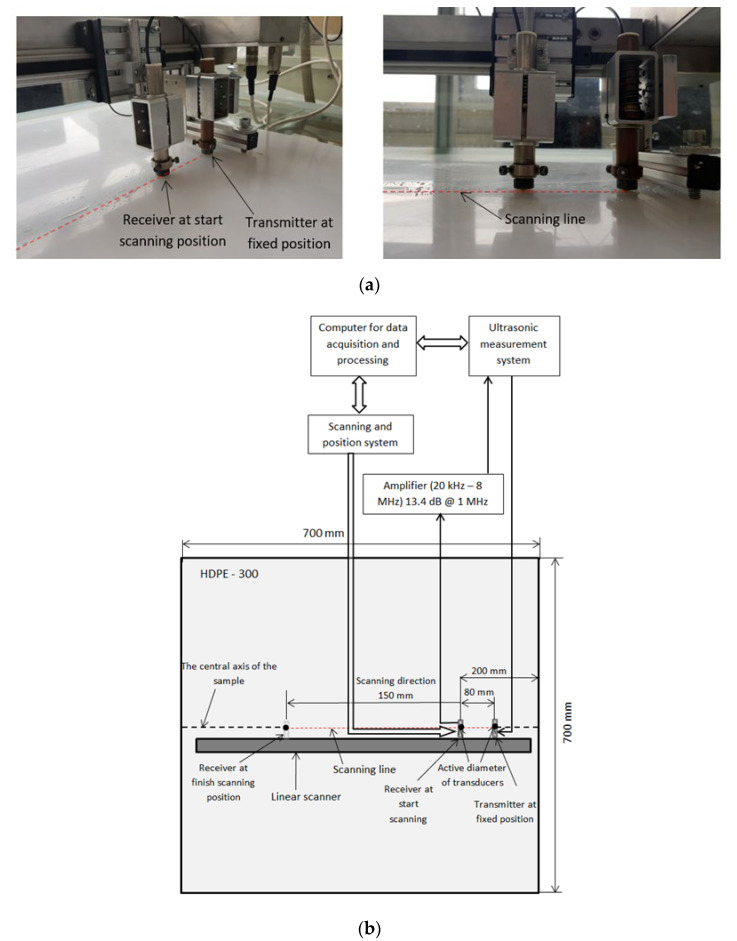
(**a**) Equipment of the measurement; (**b**) measurement set-up with linear scanner.

**Figure 9 materials-15-07249-f009:**
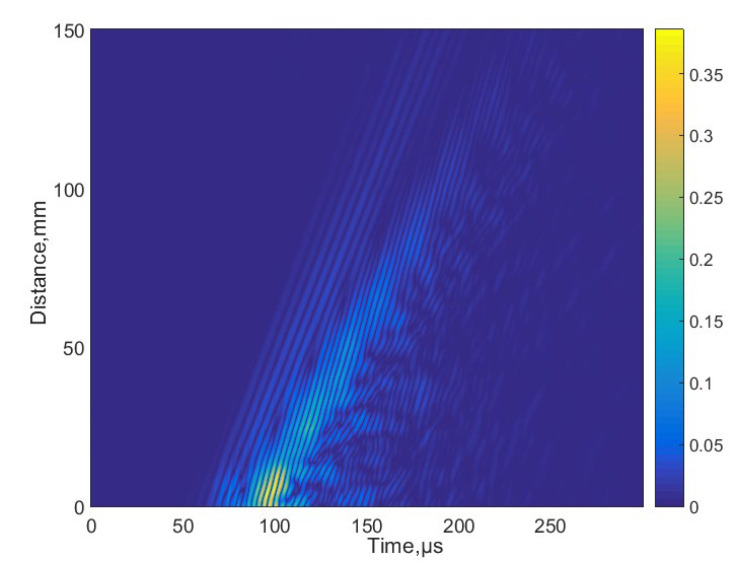
The measured B-scan obtained in the case of the contact excitation.

**Figure 10 materials-15-07249-f010:**
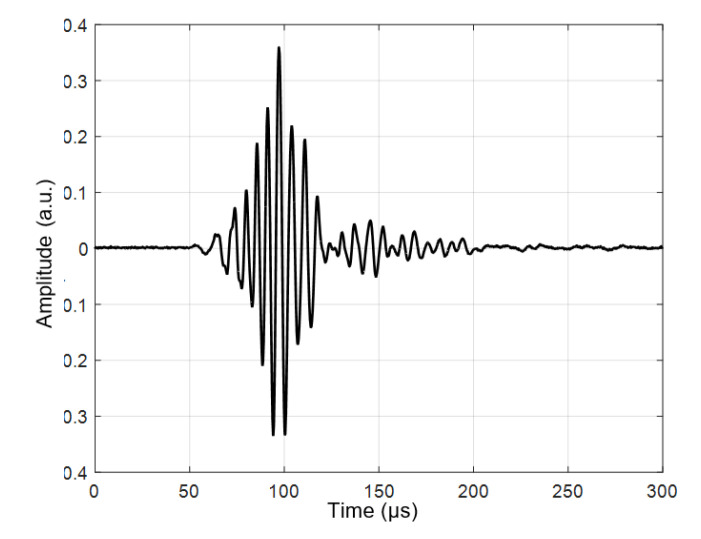
The received signal at the start position (80 mm) of the PE-300 sample. The amplitude is presented in arbitrary units (a.u.).

**Figure 11 materials-15-07249-f011:**
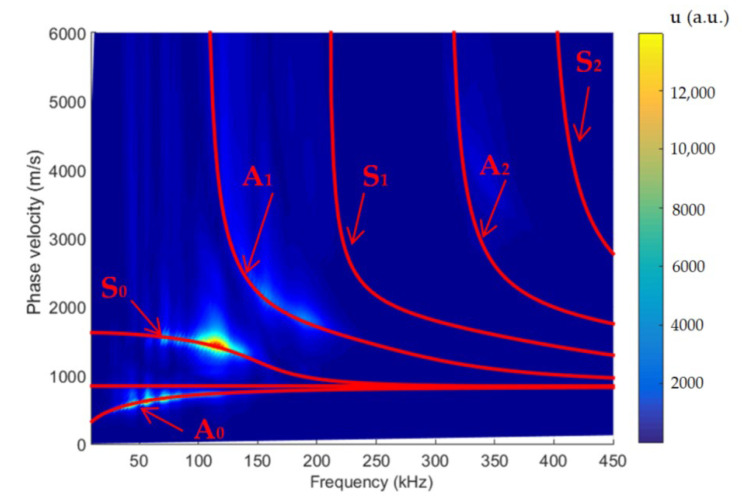
Dispersion curves obtained using SAFE (red lines) and the 2D Fourier transform FFT_2D_ of the B-scan data, PE-300 sample.

**Figure 12 materials-15-07249-f012:**
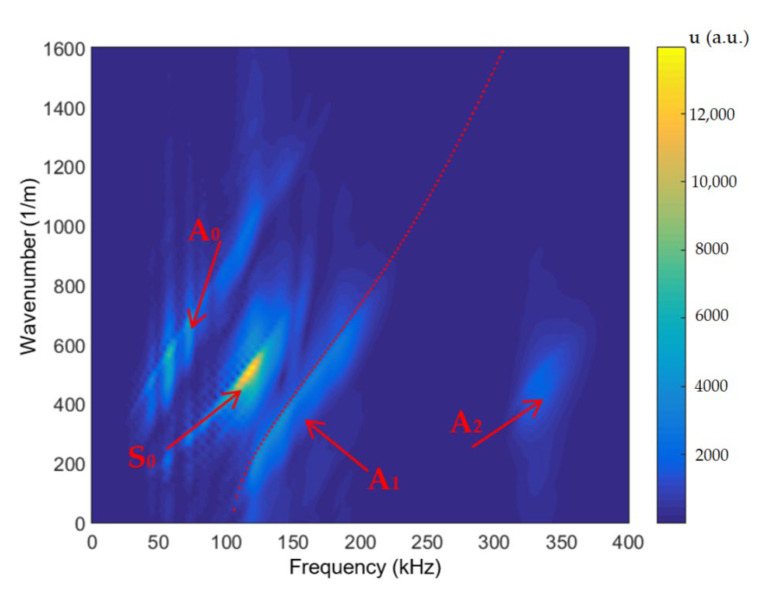
Two-dimensional spatial-temporal spectrum of the measured B-scan modes in the case of the PE-300 sample. The red dotted line is *A*_1_ mode obtained by SAFE, PE-300 sample.

**Figure 13 materials-15-07249-f013:**
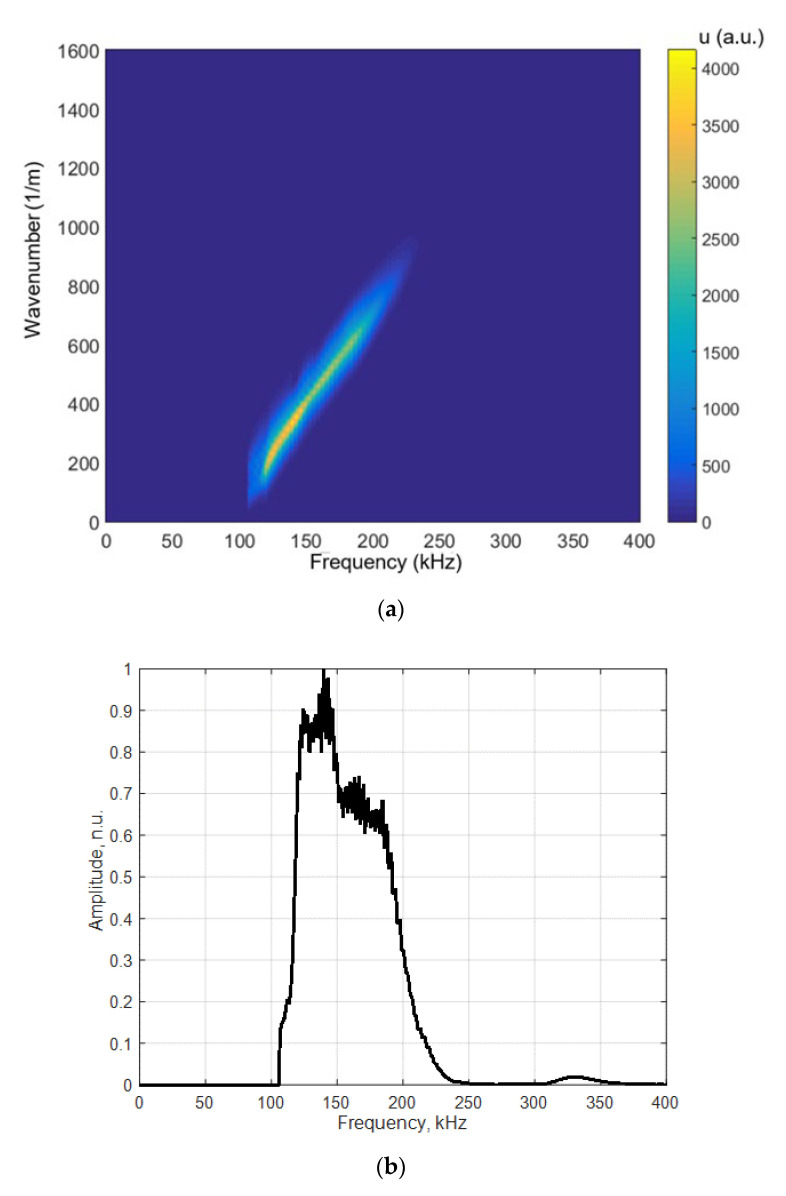
(**a**) Filtered *A*_1_ mode; (**b**) the amplitude is normalized with respect to the maximum displacement values of the filtered *A*_1_ mode (n.u.), PE-300 sample.

**Figure 14 materials-15-07249-f014:**
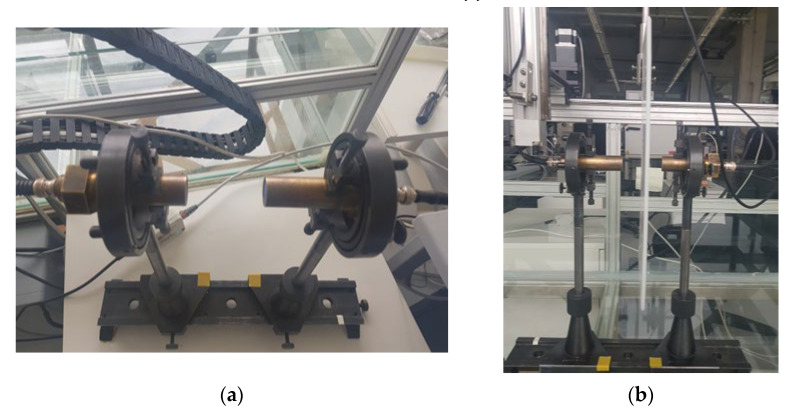
(**a**) Two air-coupled ultrasonic transducers (the transmitter and the receiver) operating “face to face” through the air gap; (**b**) air-coupled measurement with 4-axis scanner.

**Figure 15 materials-15-07249-f015:**
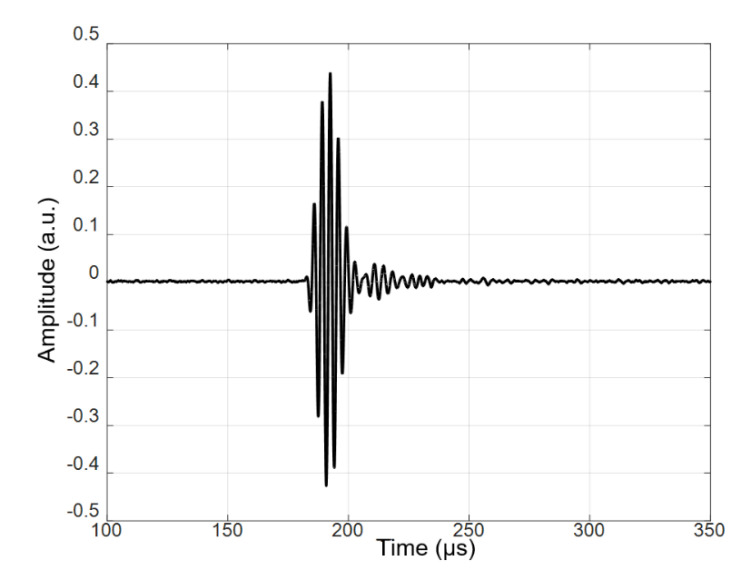
The received signal. The amplitude is presented in arbitrary units (a.u.).

**Figure 16 materials-15-07249-f016:**
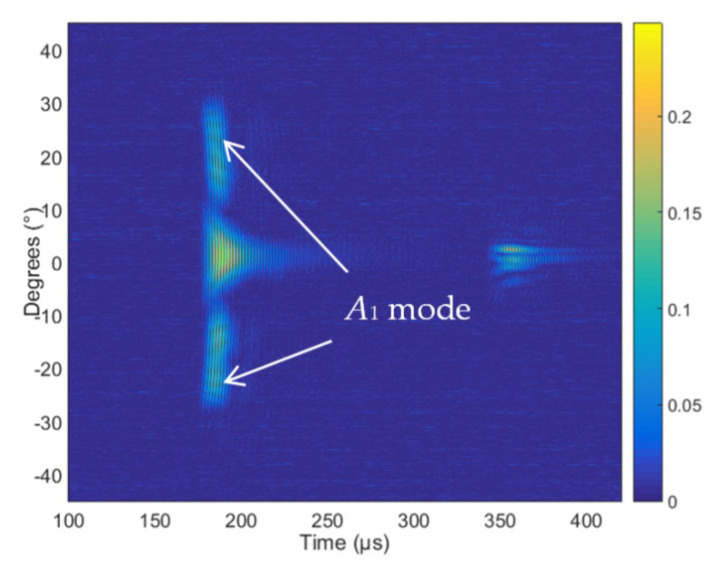
The measured B-scan obtained in the case of the air-coupled excitation of a 4 mm thick sample.

**Figure 17 materials-15-07249-f017:**
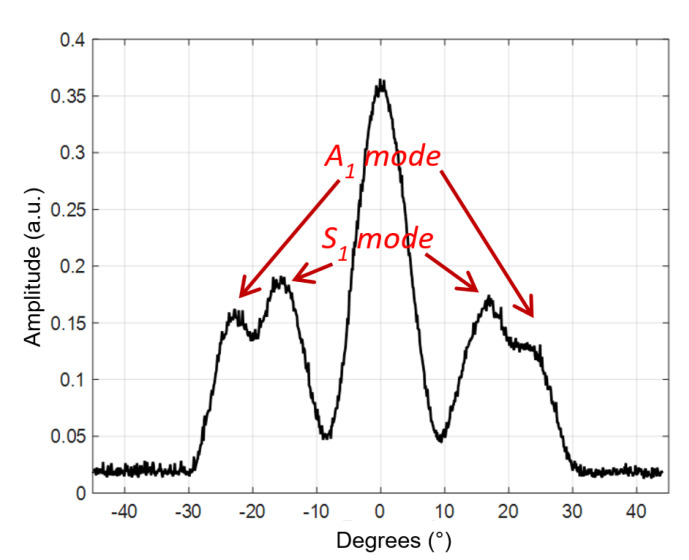
The amplitude of the measured B-scan. The amplitude of the values is presented in arbitrary units (a.u.).

**Figure 18 materials-15-07249-f018:**
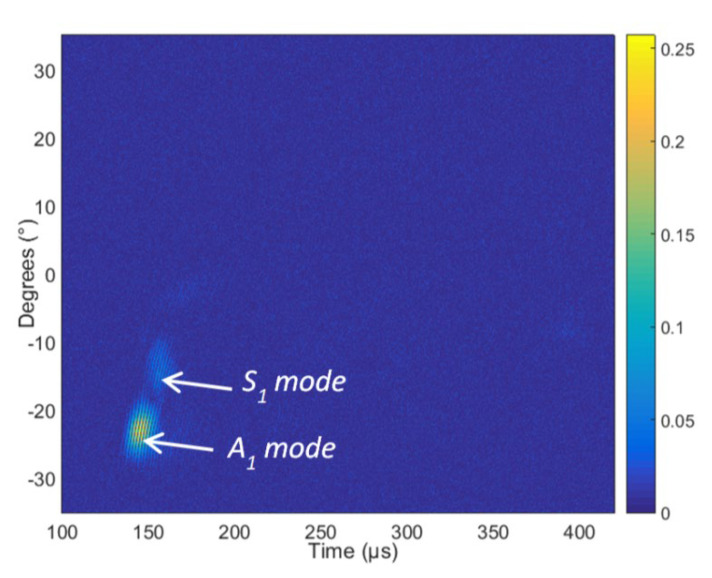
The measured B-scan obtained in the case of the air-coupled excitation.

**Figure 19 materials-15-07249-f019:**
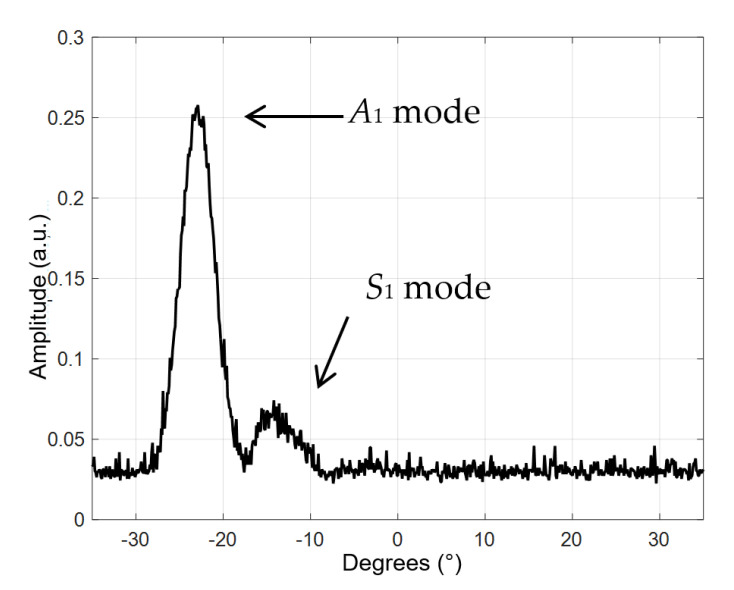
The amplitude of the measured B-scan. The amplitude of the values is presented in arbitrary units (a.u.).

**Table 1 materials-15-07249-t001:** Measured viscoelastic properties of the HDPE sample.

Parameter	Value
*C* _11_	5.2 GPa
*C* _66_	0.85 GPa

**Table 2 materials-15-07249-t002:** Calculated elastic parameters in the HDPE sample.

Parameter	Value
Density	ρ = 941 kg/m^3^
Young’s modulus	E = 2.3839 GPa
Poisson’s coefficient	ν = 0.4023

**Table 3 materials-15-07249-t003:** Measured viscoelastic properties in the GFRP sample.

Parameter	Value
*C* _11_	14.95 GPa
*C* _22_	28.96 GPa
*C* _33_	28.12 GPa
*C* _12_	6.15 GPa
*C* _13_	8.16 GPa
*C* _23_	7.00 GPa
*C* _44_	4.00 GPa
*C* _55_	3.81 GPa
*C* _66_	4.52 GPa

**Table 4 materials-15-07249-t004:** Calculated elastic parameters in the GFRP sample.

Parameter	Value
Density	ρ = 2309 kg/m^3^
Young’s modulus	E_1_ = 24.5928 GPa
	E_2_ = 12.5821 GPa
	E_3_ = 24.5044 GPa
Poisson’s coefficient	ν_12_ = 0.3833
	ν_31_ = 0.1961
	ν_13_ = 0.2063
	ν_31_ = 0.2056
	ν_23_ = 0.1618
	ν_32_ = 0.3151
Shear modulus	G_12_ = 4.00 GPa
	G_13_ = 3.81 GPa
	G_23_ = 4.52 GPa

## Data Availability

Not applicable.

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
