# Peer review of "Excitation and Reception of Higher-Order Guided Lamb Wave’s A1 and S1 Modes in Plastic and Composite Materials"

_materials, 2022, doi:10.3390/ma15207249_

Round 1

Reviewer 1 Report

This paper represented the excitation and reception of higher order guided Lamb wave in plastic and composite materials at 300 kHz. However, the research has been studied by many scholars. I think this work don't have the  enough novelty. Also, the experiments are inadequate. Authors only investigated the excitation and reception of guided wave at 300 kHz. It's better to do further investigation for samll damage detection with higher order guided wave and improve the advantage than S0 and A0 mode.

Author Response

Dear reviewer,

thank you for your comments and recommendations. We try to answer your questions. Please look at the attached file.

Reviewer 2 Report

The manuscript is concerned with analysis of the dispersion curves and the feasibility of excitation for the Lamb modes of the order higher than zero. There are three separate parts in the work presented: first, the authors reveal the elastic parameters of the materials in question (HDPE and GFRP), then use these data to calculate the dispersion curves followed by experimental validation of some calculation results obtained. The first section discloses the results obtained at the Physical Acoustics department (APY), Bordeaux, France which are then applied for calculations of the displacement patterns and the dispersion curves for some higher-order modes. The experimental section uses both contact (point-like) and non-contact (air-coupled) transducers to excite and then analyze the mode velocities by using the SAFE and 2D-FFT methods. The first excitation technique seems to be more informative (producing up to the 2d-order modes) while the ACU generates only the 1st-order symmetrical and anti-symmetrical modes.

On the whole, the results obtained are possibly of interest for the NDT community. However, their presentation has plenty of obvious drawbacks according to the comments below:

  • The Introduction creates an impression that the higher-order modes have not been studied in the literature besides SH-modes (which in reality are of minor interest for applications). The authors should go through and include in the Introduction some results obtained (and not only) in the following publications:

V.I.Anisimkin, N.V.Voronova, M.A.Zemlyanitsyn, I.E.Kuznetsova, I.I.Pyataikin, Characteristic Features of Excitation and Propagation of Acoustic Modes in Piezoelectric Plates, J.Communications and Electronics, 2013, v.58, no.10, 1004-1010.V.I.Anisimkin, N.V.Voronova, Features of Normal High-Order Acoustic Wave Generation in Thin Piezoelectric Plates, Acoustical Physics, 2020, v.66, no.1, 19-22.

  • The way the elastic parameters of the materials in question obtained in the first manuscript section (pages 3-6) remains a mystery. The reference on the papers by French researchers does not draw the light on the method and its implementation. Some “experimental“ figures on the topic do not help: the reader is puzzled by Fig. 2: why ACU transducers are put in mater? What is an “isometric material“ (page 5)? GFRP has been known to be anisotropic for years (page 6). By the way, without the data on weight/volume of fibers all calculations for GFRP do not make any sense. The accuracy of the elastic parameters presented (10-4) seems to be incredible (Table 2) and it is another reason to discuss the method used.
  • A similar comment on the SAFE method (and 2D FFT used later) applied in part 2 for calculations of the dispersion curves and remain unknown to the reader. How can one “solve the dispersion curves“ (page 7). Which direction was used in calculations for anisotropic GFRP (Fig. 4). Snell`s law was not first suggested in [30]. For A1 mode velocity determined as 1508 m/s the angle of incidence must be about 13 degrees but not 22 as stated on page 9. A sky-high accuracy in determining the wavelengths on page 9 must be validated.
  • The contact method described on page 12 is barely understandable. What are the transducers geometry and the material?  Why the specimen surface between the transmitter and receiver is covered with a couplant? That must change the velocity measured. B-scan is actually just the change in propagation distance but not a real 2D-scan (multiple times wrong usage). The legends in Figs. 11-13 must indicate the material measured. How could the sample in Fig. 14 “turned at an angle of 90 degrees “ (page 17). Fig. 15 is not the “waveform“. The calculations of the incidence angles on pages 18&19 seem like wrong.
  • The section Conclusion and discussion does not have any conclusions and discussion. Instead, it again reiterates the aims of the work and describes the methodologies used in the previous sections.
  • Some language pitfalls must also be corrected.

Author Response

(The authors gave the same response as above.)

Reviewer 3 Report

The authors investigate experimentally and numerically the excitation and propagation of higher order guided waves in HDPE and GFRP plates. Many studies for material properties identification, where similar techniques were proposed and employed [https://doi.org/10.1177/1475921713501108; https://doi.org/10.3390/ma15041307; https://doi.org/10.1121/1.5024353; https://doi.org/10.5194/jsss-5-187-2016],  were performed last years. In the present study, two approaches are compared: air-coupled excitation and non-contact sensing enables a non-destructive evaluation and characterization of plastic plates. The presented results are of certain interest, while the methods used here are not novel, so more deep conclusions are needed. Besides, higher order guided Lamb waves modes were employed in the studies mentioned above and others. Additional suggestions for the revision are given below:

- The authors have not provided details on the material properties identification procedure. The authors stated that viscoelastic properties are presented in Table 2 without any details of the characterization procedure. How the properties were evaluated?

- The title seems is too general. The authors consider only two kinds of plates and A1/S1 only from all the higher order modes, so the corresponding corrections in the title are suggested.

- Young's moduli for plates look very doubtful (24.59, 12.58 MPa). Please check carefully the table or explain these low values.

- Though Figs. 16-17 are fine, Fig. 18 does not match Fig. 19 (maxima are different). The latter must be also checked and corrected.

Author Response

(The authors gave the same response as above.)

Round 2

Reviewer 1 Report

Agree to pubulish in Materials

Author Response

Dear Reviewer,

Thank you very much for your time.

Reviewer 2 Report

Comments of the authors` response to reviewer`s comment set1:

 In general, in regard to the major reviewer comments the authors continue to refer to some publications where the information on the methodology could be clarified. My point is still different: at least core info on the methods applied (due to space limitation) for dispersion curves calculations (SAFE) and the experiments that enable to provide 10-4 accuracy in Young`s modulus and Poisson`s ratio (Tables 2, 4) must be given to make the manuscript understandable for a reader.

As for the authors` new insert on page 2, the way it has been written shows that the new references have nothing to do with the higher-order guided waves. A more rigorous analysis and more adequate comments are required.

Other questions from the reviewer`s comment set1 escaped the authors` attention and must also be worked out:

  • The reader is puzzled by Fig. 2: why ACU transducers are put in mater? What is an “isometric material“ (page 5)? GFRP has been known to be anisotropic for years (page 6). Which direction was used in calculations for anisotropic GFRP (Fig. 4). By the way, without the data on weight/volume of fibers all calculations for GFRP do not make any sense. How can one “solve the dispersion curves“ (page 7). A sky-high accuracy in determining the wavelengths on page 9 must be validated. Why the specimen surface between the transmitter and receiver is covered with a couplant (pages 12 & 13)? That must change the velocity measured. B-scan is actually just the change in propagation distance but not a real 2D-scan (multiple times wrong usage). How could the sample in Fig. 14 “turned at an angle of 90 degrees “ (page 17). The section Conclusion and discussion does not have any conclusions and discussion.

The references [17, 18] on page 3 are wrong.

The reviewer expects the replies to each of the questions above to be given separately.

Author Response

Dear Reviewer,

Thank you for the comments and sorry for not answering some questions right away. Please see the attached file and I hope I have answered all the questions this time.

Reviewer 3 Report

The authors have improved the manuscript so that it is recommended for publication.

Author Response

(The authors gave the same response as above.)
